# Learning Low-Dimensional Metrics

**Lalit Jain** *
University of Michigan
Ann Arbor, MI 48109
lalitj@umich.edu

**Blake Mason** *
University of Wisconsin
Madison, WI 53706
bmason3@wisc.edu

**Robert Nowak**
University of Wisconsin
Madison, WI 53706
rdnowak@wisc.edu

## Abstract

This paper investigates the theoretical foundations of metric learning, focused on three key questions that are not fully addressed in prior work: 1) we consider learning general low-dimensional (low-rank) metrics as well as sparse metrics; 2) we develop upper and lower (minimax) bounds on the generalization error; 3) we quantify the sample complexity of metric learning in terms of the dimension of the feature space and the dimension/rank of the underlying metric; 4) we also bound the accuracy of the learned metric relative to the underlying true generative metric. All the results involve novel mathematical approaches to the metric learning problem, and also shed new light on the special case of ordinal embedding (aka non-metric multidimensional scaling).

## 1 Low-Dimensional Metric Learning

This paper studies the problem of learning a low-dimensional Euclidean metric from comparative judgments. Specifically, consider a set of $n$ items with high-dimensional features $\boldsymbol{x}_i \in \mathbb{R}^p$ and suppose we are given a set of (possibly noisy) distance comparisons of the form

$$\text{sign}(\text{dist}(\boldsymbol{x}_i, \boldsymbol{x}_j) - \text{dist}(\boldsymbol{x}_i, \boldsymbol{x}_k)),$$

for a subset of all possible triplets of the items. Here we have in mind comparative judgments made by humans and the distance function implicitly defined according to human perceptions of similarities and differences. For example, the items could be images and the $\boldsymbol{x}_i$ could be visual features automatically extracted by a machine. Accordingly, our goal is to learn a $p \times p$ symmetric positive semi-definite (psd) matrix $\boldsymbol{K}$ such that the metric $d_{\boldsymbol{K}}(\boldsymbol{x}_i, \boldsymbol{x}_j) := (\boldsymbol{x}_i - \boldsymbol{x}_j)^T \boldsymbol{K} (\boldsymbol{x}_i - \boldsymbol{x}_j)$, where $d_{\boldsymbol{K}}(\boldsymbol{x}_i, \boldsymbol{x}_j)$ denotes the squared distance between items $i$ and $j$ with respect to a matrix $\boldsymbol{K}$, predicts the given distance comparisons as well as possible. Furthermore, it is often desired that the metric is *low-dimensional* relative to the original high-dimensional feature representation (i.e., rank$(\boldsymbol{K}) \leq d < p$). There are several motivations for this:

- Learning a high-dimensional metric may be infeasible from a limited number of comparative judgments, and encouraging a low-dimensional solution is a natural regularization.

- Cognitive scientists are often interested in visualizing human perceptual judgments (e.g., in a two-dimensional representation) and determining which features most strongly influence human perceptions. For example, educational psychologists in [1] collected comparisons between visual representations of chemical molecules in order to identify a small set of visual features that most significantly influence the judgments of beginning chemistry students.

- It is sometimes reasonable to hypothesize that a small subset of the high-dimensional features dominate the underlying metric (i.e., many irrelevant features).

- Downstream applications of the learned metric (e.g., for classification purposes) may benefit from robust, low-dimensional metrics.

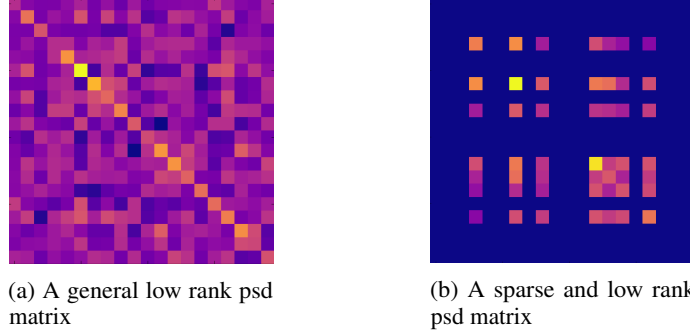

(a) A general low rank psd matrix

(b) A sparse and low rank psd matrix

Figure 1: Examples of $K$ for $p = 20$ and $d = 7$. The sparse case depicts a situation in which only some of the features are relevant to the metric.

With this in mind, several authors have proposed nuclear norm and $\ell_{1,2}$ group lasso norm regularization to encourage low-dimensional and sparse metrics as in Fig. 1b (see [2] for a review). Relative to such prior work, the contributions of this paper are three-fold:

1. We develop novel upper bounds on the generalization error and sample complexity of learning low-dimensional metrics from triplet distance comparisons. Notably, unlike previous generalization bounds, our bounds allow one to easily quantify how the feature space dimension $p$ and rank or sparsity $d < p$ of the underlying metric impacts the sample complexity.

2. We establish minimax lower bounds for learning low-rank and sparse metrics that match the upper bounds up to polylogarithmic factors, demonstrating the optimality of learning algorithms for the first time. Moreover, the upper and lower bounds demonstrate that learning sparse (and low-rank) metrics is essentially as difficult as learning a general low-rank metric. This suggests that nuclear norm regularization may be preferable in practice, since it places less restrictive assumptions on the problem.

3. We use the generalization error bounds to obtain model identification error bounds that quantify the accuracy of the learned $K$ matrix. This problem has received very little, if any, attention in the past and is crucial for interpreting the learned metrics (e.g., in cognitive science applications). This is a bit surprising, since the term "metric learning" strongly suggests accurately determining a metric, not simply learning a predictor that is parameterized by a metric.

## 1.1 Comparison with Previous Work

There is a fairly large body of work on metric learning which is nicely reviewed and summarized in the monograph [2], and we refer the reader to it for a comprehensive summary of the field. Here we discuss a few recent works most closely connected to this paper. Several authors have developed generalization error bounds for metric learning, as well as bounds for downstream applications, such as classification, based on learned metrics. To use the terminology of [2], most of the focus has been on must-link/cannot-link constraints and less on relative constraints (i.e., triplet constraints as considered in this paper). Generalization bounds based on algorithmic robustness are studied in [3], but the generality of this framework makes it difficult to quantify the sample complexity of specific cases, such as low-rank or sparse metric learning. Rademacher complexities are used to establish generalization error bounds in the must-link/cannot-link situation in [4, 5, 6], but do not consider the case of relative/triplet constraints. The sparse compositional metric learning framework of [7] does focus on relative/triplet constraints and provides generalization error bounds in terms of covering numbers. However, this work does not provide bounds on the covering numbers, making it difficult to quantify the sample complexity. To sum up, prior work does not quantify the sample complexity of metric learning based on relative/triplet constraints in terms of the intrinsic problem dimensions (i.e., dimension $p$ of the high-dimensional feature space and the dimension $d$ of the underlying metric), there is no prior work on lower bounds, and no prior work quantifying the accuracy of learned metrics themselves (i.e., only bounds on prediction errors, not model identification errors). Finally we mention that Fazel et a.l [8] also consider the recovery of sparse and low rank matrices from linear observations. Our situation is very different, our matrices are low rank because they are sparse - not sparse and simultaneously low rank as in their case.

## 2 The Metric Learning Problem

Consider $n$ known points $\boldsymbol{X} := [\boldsymbol{x}_1, \boldsymbol{x}_2, \ldots, \boldsymbol{x}_n] \in \mathbb{R}^{p \times n}$. We are interested in learning a symmetric positive semidefinite matrix $\boldsymbol{K}$ that specifies a metric on $\mathbb{R}^p$ given ordinal constraints on distances between the known points. Let $\mathcal{S}$ denote a set of triplets, where each $t = (i, j, k) \in \mathcal{S}$ is drawn uniformly at random from the full set of $n\binom{n-1}{2}$ triplets $\mathcal{T} := \{(i, j, k) : 1 \le i \ne j \ne k \le n, j < k\}$. For each triplet, we observe a $y_t \in \{\pm 1\}$ which is a noisy indication of the triplet constraint $d_{\boldsymbol{K}}(x_i, x_j) < d_{\boldsymbol{K}}(x_i, x_k)$. Specifically we assume that each $t$ has an associated probability $q_t$ of $y_t = -1$, and all $y_t$ are statistically independent.

**Objective 1:** Compute an estimate $\widehat{\boldsymbol{K}}$ from $\mathcal{S}$ that predicts triplets as well as possible.

In many instances, our triplet measurements are noisy observations of triplets from a true positive semi-definite matrix $\boldsymbol{K}^*$. In particular we assume

$$q_t > 1/2 \iff d_{\boldsymbol{K}^*}(\boldsymbol{x}_i, \boldsymbol{x}_j) < d_{\boldsymbol{K}^*}(\boldsymbol{x}_i, \boldsymbol{x}_k) .$$

We can also assume an explicit known *link function*, $f : \mathbb{R} \to [0, 1]$, so that $q_t = f(d_{\boldsymbol{K}^*}(\boldsymbol{x}_i, \boldsymbol{x}_j) - d_{\boldsymbol{K}^*}(\boldsymbol{x}_i, \boldsymbol{x}_k))$.

**Objective 2:** Assuming an explicit known link function $f$ estimate $\boldsymbol{K}^*$ from $\mathcal{S}$.

### 2.1 Definitions and Notation

Our triplet observations are nonlinear transformations of a linear function of the Gram matrix $\boldsymbol{G} := \boldsymbol{X}^T \boldsymbol{K} \boldsymbol{X}$. Indeed for any triple $t = (i, j, k)$, define

$$
\begin{aligned}
\boldsymbol{M}_t(\boldsymbol{K}) &:= d_{\boldsymbol{K}}(\boldsymbol{x}_i, \boldsymbol{x}_j) - d_{\boldsymbol{K}}(\boldsymbol{x}_i, \boldsymbol{x}_k) \\
&= \boldsymbol{x}_i^T \boldsymbol{K} \boldsymbol{x}_k + \boldsymbol{x}_k^T \boldsymbol{K} \boldsymbol{x}_i - \boldsymbol{x}_i^T \boldsymbol{K} \boldsymbol{x}_j - \boldsymbol{x}_j^T \boldsymbol{K} \boldsymbol{x}_i + \boldsymbol{x}_j^T \boldsymbol{K} \boldsymbol{x}_j - \boldsymbol{x}_k^T \boldsymbol{K} \boldsymbol{x}_k .
\end{aligned}
$$

So for every $t \in \mathcal{S}$, $y_t$ is a noisy measurement of $\text{sign}(\boldsymbol{M}_t(\boldsymbol{K}))$. This linear operator may also be expressed as a matrix

$$\boldsymbol{M}_t := \boldsymbol{x}_i \boldsymbol{x}_k^T + \boldsymbol{x}_k \boldsymbol{x}_i^T - \boldsymbol{x}_i \boldsymbol{x}_j^T - \boldsymbol{x}_j \boldsymbol{x}_i^T + \boldsymbol{x}_j \boldsymbol{x}_j^T - \boldsymbol{x}_k \boldsymbol{x}_k^T ,$$

so that $\boldsymbol{M}_t(\boldsymbol{K}) = \langle \boldsymbol{M}_t, \boldsymbol{K} \rangle = \text{Trace}(\boldsymbol{M}_t^T \boldsymbol{K})$. We will use $\boldsymbol{M}_t$ to denote the operator and associated matrix interchangeably. Ordering the elements of $\mathcal{T}$ lexicographically, we let $\mathcal{M}$ denote the linear map,

$$\mathcal{M}(\boldsymbol{K}) = (\boldsymbol{M}_t(\boldsymbol{K})| \text{ for } t \in \mathcal{T}) \in \mathbb{R}^{n\binom{n-1}{2}}$$

Given a PSD matrix $\boldsymbol{K}$ and a sample, $t \in \mathcal{S}$, we let $\ell(y_t \langle \boldsymbol{M}_t, \boldsymbol{K} \rangle)$ denote the loss of $\boldsymbol{K}$ with respect to $t$; e.g., the 0-1 loss $\mathbb{1}_{\{\text{sign}(y_t \langle \boldsymbol{M}_t, \boldsymbol{K} \rangle) \ne 1\}}$, the hinge-loss $\max\{0, 1 - y_t \langle \boldsymbol{M}_t, \boldsymbol{K} \rangle\}$, or the logistic loss $\log(1 + \exp(-y_t \langle \boldsymbol{M}_t, \boldsymbol{K} \rangle))$. Note that we insist that our losses be functions of our triplet differences $\langle \boldsymbol{M}_t, \boldsymbol{K} \rangle$. Further, note that this makes our losses invariant to rigid motions of the points $\boldsymbol{x}_i$. Other models proposed for metric learning use scale-invariant loss functions [9].

For a given loss $\ell$, we then define the empirical risk with respect to our set of observations $\mathcal{S}$ to be

$$\widehat{R}_{\mathcal{S}}(\boldsymbol{K}) := \frac{1}{|\mathcal{S}|} \sum_{t \in \mathcal{S}} \ell(y_t \langle \boldsymbol{M}_t, \boldsymbol{K} \rangle).$$

This is an unbiased estimator of the true risk $R(\boldsymbol{K}) := \mathbb{E}[\ell(y_t \langle \boldsymbol{M}_t, \boldsymbol{K} \rangle)]$ where the expectation is taken with respect to a triplet $t$ selected uniformly at random and the random value of $y_t$.

Finally, we let $\boldsymbol{I}_n$ denote the identity matrix in $\mathbb{R}^{n \times n}$, $\boldsymbol{1}_n$ the $n$-dimensional vector of all ones and $\boldsymbol{V} := \boldsymbol{I}_n - \frac{1}{n} \boldsymbol{1}_n \boldsymbol{1}_n^T$ the *centering matrix*. In particular if $\boldsymbol{X} \in \mathbb{R}^{p \times n}$ is a set of points, $\boldsymbol{X}\boldsymbol{V}$ subtracts the mean of the columns of $\boldsymbol{X}$ from each column. We say that $\boldsymbol{X}$ is centered if $\boldsymbol{X}\boldsymbol{V} = 0$, or equivalently $\boldsymbol{X}\boldsymbol{1}_n = 0$. If $\boldsymbol{G}$ is the Gram matrix of the set of points $\boldsymbol{X}$, i.e. $\boldsymbol{G} = \boldsymbol{X}^T \boldsymbol{X}$, then we say that $\boldsymbol{G}$ is centered if $\boldsymbol{X}$ is centered or if equivalently, $\boldsymbol{G}\boldsymbol{1}_n = 0$. Furthermore we use $\| \cdot \|_*$ to denote the nuclear norm, and $\| \cdot \|_{1,2}$ to denote the mixed $\ell_{1,2}$ norm of a matrix, the sum of the $\ell_2$ norms of its rows. Unless otherwise specified, we take $\| \cdot \|$ to be the standard operator norm when applied to matrices and the standard Euclidean norm when applied to vectors. Finally we define the $\boldsymbol{K}$-norm of a vector as $\|\boldsymbol{x}\|_{\boldsymbol{K}}^2 := \boldsymbol{x}^T \boldsymbol{K} \boldsymbol{x}$.

## 2.2 Sample Complexity of Learning Metrics.

In most applications, we are interested in learning a matrix $\boldsymbol{K}$ that is low-rank and positive-semidefinite. Furthermore as we will show in Theorem 2.1, such matrices can be learned using fewer samples than general psd matrices. As is common in machine learning applications, we relax the rank constraint to a nuclear norm constraint. In particular, let our constraint set be

$$\mathcal{K}_{\lambda,\gamma} = \{\boldsymbol{K} \in \mathbb{R}^{p \times p} | \boldsymbol{K} \text{ positive-semidefinite}, \|\boldsymbol{K}\|_* \leq \lambda, \max_{t \in \mathcal{T}} \langle \boldsymbol{M}_t, \boldsymbol{K} \rangle \leq \gamma\}.$$

Up to constants, a bound on $\langle \boldsymbol{M}_t, \boldsymbol{K} \rangle$ is a bound on $\boldsymbol{x}_i^T \boldsymbol{K} \boldsymbol{x}_i$. This bound along with assuming our loss function is Lipschitz, will lead to a tighter bound on the deviation of $\widehat{R}_{\mathcal{S}}(\boldsymbol{K})$ from $R(\boldsymbol{K})$ crucial in our upper bound theorem.

Let $\boldsymbol{K}^* := \min_{\boldsymbol{K} \in \mathcal{K}_{\lambda,\gamma}} R(\boldsymbol{K})$ be the true risk minimizer in this class, and let $\widehat{\boldsymbol{K}} := \min_{\boldsymbol{K} \in \mathcal{K}_{\lambda,\gamma}} \widehat{R}_{\mathcal{S}}(\boldsymbol{K})$ be the empirical risk minimizer. We achieve the following prediction error bounds for the empirical risk minimzer.

**Theorem 2.1.** *Fix $\lambda, \gamma, \delta > 0$. In addition assume that $\max_{1 \leq i \leq n} \|\boldsymbol{x}_i\|^2 = 1$. If the loss function $\ell$ is L-Lipschitz, then with probability at least $1 - \delta$*

$$R(\widehat{\boldsymbol{K}}) - R(\boldsymbol{K}^*) \leq 4L \left( \sqrt{\frac{140\lambda^2 \frac{\|\boldsymbol{X}\boldsymbol{X}^T\|}{n} \log p}{|\mathcal{S}|}} + \frac{2 \log p}{|\mathcal{S}|} \right) + \sqrt{\frac{2L^2\gamma^2 \log 2/\delta}{|\mathcal{S}|}}$$

Note that past generalization error bounds in the metric learning literature have failed to quantify the precise dependence on observation noise, dimension, rank, and our features $\boldsymbol{X}$. Consider the fact that a $p \times p$ matrix with rank $d$ has $O(dp)$ degrees of freedom. With that in mind, one expects the sample complexity to be also roughly $O(dp)$. We next show that this intuition is correct if the original representation $\boldsymbol{X}$ is isotropic (i.e., has no preferred direction).

**The Isotropic Case.** Suppose that $\boldsymbol{x}_1, \cdots, \boldsymbol{x}_n, n > p$, are drawn independently from the isotropic Gaussian $\mathcal{N}(\boldsymbol{0}, \frac{1}{p}\boldsymbol{I})$. Furthermore, suppose that $\boldsymbol{K}^* = \frac{p}{\sqrt{d}}\boldsymbol{U}\boldsymbol{U}^T$ with $\boldsymbol{U} \in \mathbb{R}^{p \times d}$ is a generic (dense) orthogonal matrix with unit norm columns. The factor $\frac{p}{\sqrt{d}}$ is simply the scaling needed so that the average magnitude of the entries in $\boldsymbol{K}^*$ is a constant, independent of the dimensions $p$ and $d$. In this case, $\text{rank}(\boldsymbol{K}^*) = d$ and $\|\boldsymbol{K}^*\|_F = \text{trace}(\boldsymbol{U}^T\boldsymbol{U}) = p$. These two facts imply that the tightest bound on the nuclear norm of $\boldsymbol{K}^*$ is $\|\boldsymbol{K}^*\|_* \leq p\sqrt{d}$. Thus, we take $\lambda = p\sqrt{d}$ for the nuclear norm constraint. Now let $\boldsymbol{z}_i = \sqrt{\frac{p}{\sqrt{d}}}\boldsymbol{U}^T\boldsymbol{x}_i \sim N(\boldsymbol{0}, \boldsymbol{I}_d)$ and note that $\|\boldsymbol{x}_i\|_{\boldsymbol{K}}^2 = \|\boldsymbol{z}_i\|^2 \sim \chi_d^2$. Therefore, $\mathbb{E}\|\boldsymbol{x}_i\|_{\boldsymbol{K}}^2 = d$ and it follows from standard concentration bounds that with large probability $\max_i \|\boldsymbol{x}_i\|_{\boldsymbol{K}}^2 \leq 5d \log n =: \gamma$ see [10]. Also, because the $\boldsymbol{x}_i \sim \mathcal{N}(\boldsymbol{0}, \frac{1}{p}\boldsymbol{I})$ it follows that if $n > p \log p$, say, then with large probability $\|\boldsymbol{X}\boldsymbol{X}^T\| \leq 5n/p$. We now plug these calculations into Theorem 2.1 to obtain the following corollary.

**Corollary 2.1.1** (Sample complexity for isotropic points). *Fix $\delta > 0$, set $\lambda = p\sqrt{d}$, and assume that $\|\boldsymbol{X}\boldsymbol{X}^T\| = O(n/p)$ and $\gamma := \max_i \|\boldsymbol{x}_i\|_K^2 = O(d \log n)$. Then for a generic $\boldsymbol{K}^* \in \mathcal{K}_{\lambda,\gamma}$, as constructed above, with probability at least $1 - \delta$,*

$$R(\widehat{\boldsymbol{K}}) - R(\boldsymbol{K}^*) = O\left( \sqrt{\frac{dp(\log p + \log^2 n)}{|\mathcal{S}|}} \right)$$

This bound agrees with the intuition that the sample complexity should grow roughly like $dp$, the degrees of freedom on $\boldsymbol{K}^*$. Moreover, our minimax lower bound in Theorem 2.3 below shows that, ignoring logarithmic factors, the general upper bound in Theorem 2.1 is unimprovable in general.

Beyond low rank metrics, in many applications it is reasonable to assume that only a few of the features are salient and should be given nonzero weight. Such a metric may be learned by insisting $\boldsymbol{K}$ to be row sparse in addition to being low rank. Whereas learning a low rank $\boldsymbol{K}$ assumes that distance is well represented in a low dimensional subspace, a row sparse (and hence low rank) $\boldsymbol{K}$ defines a metric using only a subset of the features. Figure 1 gives a comparison of a low rank versus a low rank and sparse matrix $\boldsymbol{K}$.

Analogous to the convex relaxation of rank by the nuclear norm, it is common to relax row sparsity by using the mixed $\ell_{1,2}$ norm. In fact, the geometry of the $\ell_{1,2}$ and nuclear norm balls are tightly related as the following lemma shows.

**Lemma 2.2.** *For a symmetric positive semi-definite matrix $\boldsymbol{K} \in \mathbb{R}^{p \times p}$, $\|\boldsymbol{K}\|_* \leq \|\boldsymbol{K}\|_{1,2}$.*

*Proof.* $\|\boldsymbol{K}\|_{1,2} = \sum_{i=1}^{p} \sqrt{\sum_{j=1}^{p} \boldsymbol{K}_{i,j}^2} \geq \sum_{i=1}^{p} \boldsymbol{K}_{i,i} = \mathrm{Trace}(\boldsymbol{K}) = \sum_{i=1}^{p} \lambda_i(\boldsymbol{K}) = \|\boldsymbol{K}\|_*$ $\qquad\square$

This implies that the $\ell_{1,2}$ ball of a given radius is contained inside the nuclear norm ball of the same radius. In particular, it is reasonable to assume that it is easier to learn a $\boldsymbol{K}$ that is sparse in addition to being low rank. Surprisingly, however, the following minimax bound shows that this is not necessarily the case.

To make this more precise, we will consider optimization over the set

$$\mathcal{K}'_{\lambda,\gamma} = \{\boldsymbol{K} \in \mathbb{R}^{p \times p} | \boldsymbol{K} \text{ positive-semidefinite}, \|\boldsymbol{K}\|_{1,2} \leq \lambda, \max_{t \in \mathcal{T}} \langle \boldsymbol{M}_t, \boldsymbol{K} \rangle \leq \gamma\}.$$

Furthermore, we must specify the way in which our data could be generated from noisy triplet observations of a fixed $\boldsymbol{K}^*$. To this end, assume the existence of a *link function* $f : \mathbb{R} \to [0, 1]$ so that $q_t = \mathbb{P}(y_t = -1) = f(\boldsymbol{M}_t(\boldsymbol{K}^*))$ governs the observations. There is a natural associated logarithmic loss function $\ell_f$ corresponding to the log-likelihood, where the loss of an arbitrary $\boldsymbol{K}$ is

$$\ell_f(y_t \langle \boldsymbol{M}_t, \boldsymbol{K} \rangle) = \mathbb{1}_{\{y_t = -1\}} \log \frac{1}{f(\langle \boldsymbol{M}_t, \boldsymbol{K} \rangle)} + \mathbb{1}_{\{y_t = 1\}} \log \frac{1}{1 - f(\langle \boldsymbol{M}_t, \boldsymbol{K} \rangle)}$$

**Theorem 2.3.** *Choose a link function $f$ and let $\ell_f$ be the associated logarithmic loss. For $p$ sufficiently large, then there exists a choice of $\gamma$, $\lambda$, $\boldsymbol{X}$, and $|\mathcal{S}|$ such that*

$$\inf_{\widehat{\boldsymbol{K}}} \sup_{\boldsymbol{K} \in \mathcal{K}'_{\lambda,\gamma}} \mathbb{E}[R(\widehat{\boldsymbol{K}})] - R(\boldsymbol{K}) \geq C \sqrt{\frac{C_1^3 \ln 4}{2} \frac{\lambda^2 \frac{\|\boldsymbol{X}\boldsymbol{X}^T\|}{n}}{|\mathcal{S}|}}$$

*where $C = \frac{C_f^2}{32} \sqrt{\frac{\inf_{|x| \leq \gamma} f(x)(1 - f(x))}{\sup_{|\nu| \leq \gamma} f'(\nu)^2}}$ with $C_f = \inf_{|x| \leq \gamma} f'(x)$, $C_1$ is an absolute constant, and the infimum is taken over all estimators $\widehat{\boldsymbol{K}}$ of $\boldsymbol{K}$ from $|\mathcal{S}|$ samples.*

Importantly, up to polylogarithmic factors and constants, our minimax lower bound over the $\ell_{1,2}$ ball matches the upper bound over the nuclear norm ball given in Theorem 2.1. *In particular, in the worst case, learning a sparse and low rank matrix $\boldsymbol{K}$ is no easier than learning a $\boldsymbol{K}$ that is simply low rank.* However in many realistic cases, a slight performance gain is seen from optimizing over the $\ell_{1,2}$ ball when $\boldsymbol{K}^*$ is row sparse, while optimizing over the nuclear norm ball does better when $\boldsymbol{K}^*$ is dense. We show examples of this in the Section 3. The proof is given in the supplementary materials.

Note that if $\gamma$ is in a bounded range, then the constant $C$ has little effect. For the case that $f$ is the logistic function, $C_f \geq \frac{1}{4} e^{-y_t \langle \boldsymbol{M}_t, \boldsymbol{K} \rangle} \geq \frac{1}{4} e^{-\gamma}$. Likewise, the term under the root will be also be bounded for $\gamma$ in a constant range. The terms in the constant $C$ arise when translating from risk and a KL-divergence to squared distance and reflects the noise in the problem.

## 2.3 Sample Complexity Bounds for Identification

Under a general loss function and arbitrary $\boldsymbol{K}^*$, we can not hope to convert our prediction error bounds into a recovery statement. However in this section we will show that as long as $\boldsymbol{K}^*$ is low rank, and if we choose the loss function to be the log loss $\ell_f$ of a given link function $f$ as defined prior to the statement of Theorem 2.3, recovery is possible. Firstly, note that under these assumptions we have an explicit formula for the risk,

$$R(\boldsymbol{K}) = \frac{1}{|\mathcal{T}|} \sum_{t \in \mathcal{T}} f(\langle \boldsymbol{M}_t, \boldsymbol{K}^* \rangle) \log \frac{1}{f(\langle \boldsymbol{M}_t, \boldsymbol{K} \rangle)} + (1 - f(\langle \boldsymbol{M}_t, \boldsymbol{K}^* \rangle)) \log \frac{1}{1 - f(\langle \boldsymbol{M}_t, \boldsymbol{K} \rangle)}$$

and

$$R(\mathbf{K}) - R(\mathbf{K}^*) = \frac{1}{|\mathcal{T}|} \sum_{t \in \mathcal{T}} KL(f(\langle \mathbf{M}_t, \mathbf{K}^* \rangle) || f(\langle \mathbf{M}_t, \mathbf{K} \rangle)).$$

The following theorem shows that if the excess risk is small, i.e. $R(\widehat{\mathbf{K}})$ approximates $R(\mathbf{K}^*)$ well, then $\mathcal{M}(\widehat{\mathbf{K}})$ approximates $\mathcal{M}(\mathbf{K}^*)$ well. The proof, given in the supplementary materials, uses standard Taylor series arguments to show the KL-divergence is bounded below by squared-distance.

**Lemma 2.4.** *Let $C_f = \inf_{|x| \leq \gamma} f'(x)$. Then for any $\mathbf{K} \in \mathbf{K}_{\lambda,\gamma}$,*

$$\frac{2C_f^2}{|\mathcal{T}|} \|\mathcal{M}(\mathbf{K}) - \mathcal{M}(\mathbf{K}^*)\|^2 \leq R(\mathbf{K}) - R(\mathbf{K}^*).$$

The following may give us hope that recovering $\mathbf{K}^*$ from $\mathcal{M}(\mathbf{K}^*)$ is trivial, but the linear operator $\mathcal{M}$ is non-invertible in general, as we discuss next. To see why, we must consider a more general class of operators defined on Gram matrices. Given a symmetric matrix $\mathbf{G}$, define the operator $\mathbf{L}_t$ by

$$\mathbf{L}_t(\mathbf{G}) = 2\mathbf{G}_{ik} - 2\mathbf{G}_{ij} + \mathbf{G}_{jj} - \mathbf{G}_{kk}$$

If $\mathbf{G} = \mathbf{X}^T \mathbf{K} \mathbf{X}$ then $\mathbf{L}_t(\mathbf{G}) = \mathbf{M}_t(\mathbf{K})$, and more so $\mathbf{M}_t = \mathbf{X} \mathbf{L}_t \mathbf{X}^T$. Analogous to $\mathcal{M}$, we will combine the $\mathbf{L}_t$ operators into a single operator $\mathcal{L}$,

$$\mathcal{L}(\mathbf{G}) = (\mathbf{L}_t(\mathbf{G})| \text{ for } t \in \mathcal{T}) \in \mathbb{R}^{n\binom{n-1}{2}}.$$

**Lemma 2.5.** *The null space of $\mathcal{L}$ is one dimensional, spanned by $\mathbf{V} = \mathbf{I}_n - \frac{1}{n}\mathbf{1}_n\mathbf{1}_n^T$.*

The proof is contained in the supplementary materials. In particular we see that $\mathcal{M}$ is not invertible in general, adding a serious complication to our argument. However $\mathcal{L}$ is still invertible on the subset of centered symmetric matrices orthogonal to $\mathbf{V}$, a fact that we will now exploit. We can decompose $\mathbf{G}$ into $\mathbf{V}$ and a component orthogonal to $\mathbf{V}$ denoted $\mathbf{H}$,

$$\mathbf{G} = \mathbf{H} + \sigma_{\mathbf{G}} \mathbf{V}$$

where $\sigma_{\mathbf{G}} := \frac{\langle \mathbf{G}, \mathbf{V} \rangle}{\|\mathbf{V}\|_F^2}$, and under the assumption that $\mathbf{G}$ is centered, $\sigma_{\mathbf{G}} = \frac{\|\mathbf{G}\|_*}{n-1}$. Remarkably, the following lemma tells us that a *non-linear* function of $\mathbf{H}$ uniquely determines $\mathbf{G}$.

**Lemma 2.6.** *If $n > d + 1$, and $\mathbf{G}$ is rank $d$ and centered, then $-\sigma_{\mathbf{G}}$ is an eigenvalue of $\mathbf{H}$ with multiplicity $n - d - 1$. In addition, given another Gram matrix $\mathbf{G}'$ of rank $d'$, $\sigma_{\mathbf{G}'} - \sigma_{\mathbf{G}}$ is an eigenvalue of $\mathbf{H} - \mathbf{H}'$ with multiplicity at least $n - d - d' - 1$.*

*Proof.* Since $\mathbf{G}$ is centered, $\mathbf{1}_n \in \ker \mathbf{G}$, and in particular $\dim(\mathbf{1}_n^{\perp} \cap \ker \mathbf{G}) = n - d - 1$. If $x \in \mathbf{1}_n^{\perp} \cap \ker \mathbf{G}$, then

$$\mathbf{G}x = \mathbf{H}x + \sigma_{\mathbf{G}}\mathbf{V}x \Rightarrow \mathbf{H}x = -\sigma_{\mathbf{G}}x.$$

For the second statement, notice that $\dim(\mathbf{1}_n^{\perp} \cap \ker \mathbf{G} - \mathbf{G}') \geq n - d - d' - 1$. A similar argument then applies. $\square$

If $n > 2d$, then the multiplicity of the eigenvalue $-\sigma_{\mathbf{G}}$ is at least $n/2$. So we can trivially identify it from the spectrum of $\mathbf{H}$. This gives us a *non-linear* way to recover $\mathbf{G}$ from $\mathbf{H}$.

Now we can return to the task of recovering $\mathbf{K}^*$ from $\mathcal{M}(\widehat{\mathbf{K}})$. Indeed the above lemma implies that $\mathbf{G}^*$ (and hence $\mathbf{K}^*$ if $\mathbf{X}$ is full rank) can be recovered from $\mathbf{H}^*$ by computing an eigenvalue of $\mathbf{H}^*$. However $\mathbf{H}^*$ is recoverable from $\mathcal{L}(\mathbf{H}^*)$, which is itself well approximated by $\mathcal{L}(\widehat{\mathbf{H}}) = \mathcal{M}(\widehat{\mathbf{K}})$. The proof of the following theorem makes this argument precise.

**Theorem 2.7.** *Assume that $\mathbf{K}^*$ is rank $d$, $\widehat{\mathbf{K}}$ is rank $d'$, $n > d + d' + 1$, $\mathbf{X}$ is rank $p$ and $\mathbf{X}^T\mathbf{K}^*\mathbf{X}$ and $\mathbf{X}^T\widehat{\mathbf{K}}\mathbf{X}$ are all centered. Let $C_{d,d'} = \left(1 + \frac{n-1}{(n-d-d'-1)}\right)$. Then with probability at least $1 - \delta$,*

$$\frac{n\sigma_{min}(\mathbf{X}\mathbf{X}^T)^2}{|\mathcal{T}|} \|\widehat{\mathbf{K}} - \mathbf{K}^*\|_F^2 \leq \frac{2LC_{d,d'}}{C_f^2} \left[ \left( \sqrt{\frac{140\lambda^2 \frac{\|\mathbf{X}\mathbf{X}^T\|}{n} \log p}{|\mathcal{S}|}} + \frac{2\log p}{|\mathcal{S}|} \right) + \sqrt{\frac{2L^2\gamma^2 \log \frac{2}{\delta}}{|\mathcal{S}|}} \right]$$

*where $\sigma_{min}(\mathbf{X}\mathbf{X}^T)$ is the smallest eigenvalue of $\mathbf{X}\mathbf{X}^T$.*

The proof, given in the supplementary materials, relies on two key components, Lemma 2.6 and a type of *restricted isometry property* for $\mathcal{M}$ on $\boldsymbol{V}^{\perp}$. Our proof technique is a streamlined and more general approach similar to that used in the special case of ordinal embedding. In fact, our new bound improves on the recovery bound given in [11] for ordinal embedding.

We have several remarks about the bound in the theorem. If $\boldsymbol{X}$ is well conditioned, e.g. isotropic, then $\sigma_{\min}(\boldsymbol{X}\boldsymbol{X}^T) \approx \frac{n}{p}$. In that case $\frac{n\sigma_{\min}(\boldsymbol{X}\boldsymbol{X}^T)^2}{|\mathcal{T}|} \approx \frac{1}{p^2}$, so the left hand side is the average squared error of the recovery. In most applications the rank of the empirical risk minimizer $\widehat{\boldsymbol{K}}$ is approximately equal to the rank of $\boldsymbol{K}^*$, i.e. $d \approx d'$. Note that If $d + d' \leq \frac{1}{2}(n-1)$ then $C_{d,d'} \leq 3$. Finally, the assumption that $\boldsymbol{X}^T\boldsymbol{K}^*\boldsymbol{X}$ are centered can be guaranteed by centering $\boldsymbol{X}$, which has no impact on the triplet differences $\langle \boldsymbol{M}_t, \boldsymbol{K}^* \rangle$, or insisting that $\boldsymbol{K}^*$ is centered. As mentioned above $C_f$ will be have little effect assuming that our measurements $\langle \boldsymbol{M}_t, \boldsymbol{K} \rangle$ are bounded.

## 2.4 Applications to Ordinal Embedding

In the ordinal embedding setting, there are a set of items with unknown locations, $\boldsymbol{z}_1, \cdots, \boldsymbol{z}_n \in \mathbb{R}^d$ and a set of triplet observations $\mathcal{S}$ where as in the metric learning case observing $y_t = -1$, for a triplet $t = (i,j,k)$ is indicative of the $\|\boldsymbol{z}_i - \boldsymbol{z}_j\|^2 \leq \|\boldsymbol{z}_i - \boldsymbol{z}_k\|^2$, i.e. item $i$ is closer to $j$ than $k$. The goal is to recover the $\boldsymbol{z}_i$'s, up to rigid motions, by recovering their Gram matrix $\boldsymbol{G}^*$ from these comparisons. Ordinal embedding case reduces to metric learning through the following observation. Consider the case when $n = p$ and $\boldsymbol{X} = \boldsymbol{I}_p$, i.e. the $\boldsymbol{x}_i$ are standard basis vectors. Letting $\boldsymbol{K}^* = \boldsymbol{G}^*$, we see that $\|\boldsymbol{x}_i - \boldsymbol{x}_j\|_{\boldsymbol{K}}^2 = \|\boldsymbol{z}_i - \boldsymbol{z}_j\|^2$. So in particular, $\boldsymbol{L}_t = \boldsymbol{M}_t$ for each triple $t$, and observations are exactly comparative distance judgements. Our results then apply, and extend previous work on sample complexity in the ordinal embedding setting given in [11]. In particular, though Theorem 5 in [11] provides a consistency guarantee that the empirical risk minimizer $\widehat{\boldsymbol{G}}$ will converge to $\boldsymbol{G}^*$, they do not provide a convergence rate. We resolve this issue now.

In their work, it is assumed that $\|\boldsymbol{z}_i\|^2 \leq \gamma$ and $\|\boldsymbol{G}\|_* \leq \sqrt{d}n\gamma$. In particular, sample complexity results of the form $O(dn\gamma \log n)$ are obtained. However, these results are trivial in the following sense, if $\|\boldsymbol{z}_i\|^2 \leq \gamma$ then $\|\boldsymbol{G}\|_* \leq \gamma n$, and their results (as well as our upper bound) implies that true sample complexity is significantly smaller, namely $O(\gamma n \log n)$ which is independent of the ambient dimension $d$. As before, assume an explicit link function $f$ with Lipschitz constant $L$, so the samples are noisy observations governed by $\boldsymbol{G}^*$, and take the loss to be the logarithmic loss associated to $f$.

We obtain the following improved recovery bound in this case. The proof is immediate from Theorem 2.7.

**Corollary 2.7.1.** *Let $\boldsymbol{G}^*$ be the Gram matrix of $n$ centered points in $d$ dimensions with $\|\boldsymbol{G}^*\|_F^2 = \frac{\gamma^2 n^2}{d}$. Let $\widehat{\boldsymbol{G}} = \min_{\|\boldsymbol{G}\|_* \leq \gamma n, \|\boldsymbol{G}\|_\infty \leq \gamma} R_{\mathcal{S}}(\boldsymbol{G})$ and assume that $\widehat{\boldsymbol{G}}$ is rank $d$, with $n > 2d+1$. Then,*

$$\frac{\|\widehat{\boldsymbol{G}} - \boldsymbol{G}^*\|_F^2}{n^2} = O\left( \frac{LC_{d,d}}{C_f^2} \sqrt{\frac{\gamma n \log n}{|\mathcal{S}|}} \right)$$

## 3 Experiments

To validate our complexity and recovery guarantees, we ran the following simulations. We generate $\boldsymbol{x}_1, \cdots, \boldsymbol{x}_n \overset{\text{iid}}{\sim} \mathcal{N}(\boldsymbol{0}, \frac{1}{p}\boldsymbol{I})$, with $n = 200$, and $\boldsymbol{K}^* = \frac{p}{\sqrt{d}}\boldsymbol{U}\boldsymbol{U}^T$ for a random orthogonal matrix $\boldsymbol{U} \in \mathbb{R}^{p \times d}$ with unit norm columns. In Figure 2a, $\boldsymbol{K}^*$ has $d$ nonzero rows/columns. In Figure 2b, $\boldsymbol{K}^*$ is a dense rank-$d$ matrix. We compare the performance of nuclear norm and $\ell_{1,2}$ regularization in each setting against an unconstrained baseline where we only enforce that $\boldsymbol{K}$ be psd. Given a fixed number of samples, each method is compared in terms of the relative excess risk, $\frac{R(\widehat{\boldsymbol{K}}) - R(\boldsymbol{K}^*)}{R(\boldsymbol{K}^*)}$, and the relative squared recovery error, $\frac{\|\widehat{\boldsymbol{K}} - \boldsymbol{K}^*\|_F^2}{\|\boldsymbol{K}^*\|_F^2}$, averaged over 20 trials. The y-axes of both plots have been trimmed for readability.

In the case that $\boldsymbol{K}^*$ is sparse, $\ell_{1,2}$ regularization outperforms nuclear norm regularization. However, in the case of dense low rank matrices, nuclear norm reularization is superior. Notably, as expected from our upper and lower bounds, the performances of the two approaches seem to be within constant

factors of each other. Therefore, unless there is strong reason to believe that the underlying $K^*$ is sparse, nuclear norm regularization achieves comparable performance with a less restrictive modeling assumption. Furthermore, in the two settings, both the nuclear norm and $\ell_{1,2}$ constrained methods outperform the unconstrained baseline, especially in the case where $K^*$ is low rank and sparse.

To empirically validate our sample complexity results, we compute the number of samples averaged over 20 runs to achieve a relative excess risk of less than 0.1 in Figure 3. First, we fix $p = 100$ and increment $d$ from 1 to 10. Then we fix $d = 10$ and increment $p$ from 10 to 100 to clearly show the linear dependence of the sample complexity on $d$ and $p$ as demonstrated in Corollary 2.1.1. To our knowledge, these are the first results quantifying the sample complexity in terms of the number of features, $p$, and the embedding dimension, $d$.

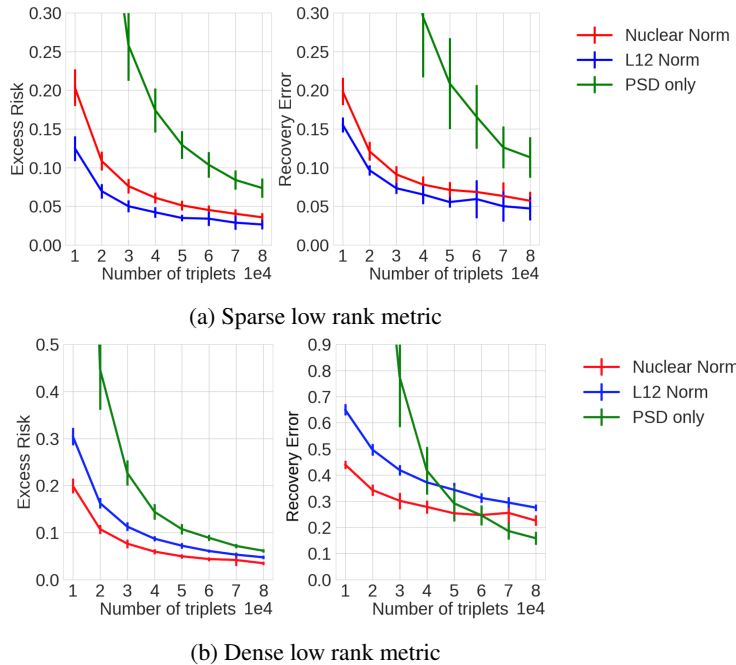

(a) Sparse low rank metric

(b) Dense low rank metric

Figure 2: $\ell_{1,2}$ and nuclear norm regularization performance

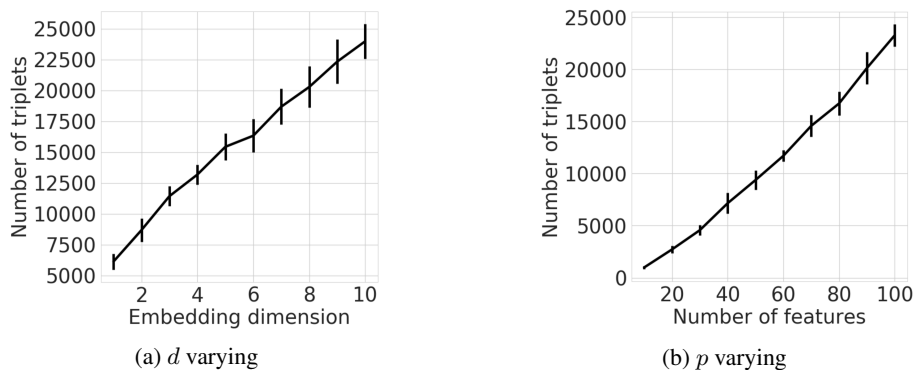

(a) $d$ varying

(b) $p$ varying

Figure 3: Number of samples to achieve relative excess risk $< 0.1$

**Acknowledgments** This work was partially supported by the NSF grants CCF-1218189 and IIS-1623605

## Footnotes

* Authors contributed equally to this paper and are listed alphabetically.

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
