[Supplementary Material]

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

{\boldsymbol{K}})$ approximates $R(\boldsymbol{K}^*)$ well, then $\mathcal{M}(\widehat{\boldsymbol{K}})$ approximates $\mathcal{M}(\boldsymbol{K}^*)$ well. The proof, given in the supplementary materials, uses standard Taylor series arguments to show the KL-divergence is bounded below by squared-distance.

**Lemma 2.4.** *Let $C_f = \inf_{|x| \le \gamma} f'(x)$. Then for any $\boldsymbol{K} \in \boldsymbol{K}_{\lambda,\gamma}$,*

$$\frac{2C_f^2}{|\mathcal{T}|} \|\mathcal{M}(\boldsymbol{K}) - \mathcal{M}(\boldsymbol{K}^*)\|^2 \le R(\boldsymbol{K}) - R(\boldsymbol{K}^*).$$

The following may give us hope that recovering $\boldsymbol{K}^*$ from $\mathcal{M}(\boldsymbol{K}^*)$ is trivial, but the linear operator $\mathcal{M}$ is non-invertible in general, as we discuss next. To see why, we must consider a more general class of operators defined on Gram matrices. Given a symmetric matrix $\boldsymbol{G}$, define the operator $\boldsymbol{L}_t$ by

$$\boldsymbol{L}_t(\boldsymbol{G}) = 2\boldsymbol{G}_{ik} - 2\boldsymbol{G}_{ij} + \boldsymbol{G}_{jj} - \boldsymbol{G}_{kk}$$

If $\boldsymbol{G} = \boldsymbol{X}^T \boldsymbol{K} \boldsymbol{X}$ then $\boldsymbol{L}_t(\boldsymbol{G}) = \boldsymbol{M}_t(\boldsymbol{K})$, and more so $\boldsymbol{M}_t = \boldsymbol{X} \boldsymbol{L}_t \boldsymbol{X}^T$. Analogous to $\mathcal{M}$, we will combine the $\boldsymbol{L}_t$ operators into a single operator $\mathcal{L}$,

$$\mathcal{L}(\boldsymbol{G}) = (\boldsymbol{L}_t(\boldsymbol{G})| \text{ for } t \in \mathcal{T}) \in \mathbb{R}^{n\binom{n-1}{2}}.$$

**Lemma 2.5.** *The null space of $\mathcal{L}$ is one dimensional, spanned by $\boldsymbol{V} = \boldsymbol{I}_n - \frac{1}{n}\boldsymbol{1}_n\boldsymbol{1}_n^T$.*

The proof is contained in the supplementary materials. In particular we see that $\mathcal{M}$ is not invertible in general, adding a serious complication to our argument. However $\mathcal{L}$ is still invertible on the subset of centered symmetric matrices orthogonal to $\boldsymbol{V}$, a fact that we will now exploit. We can decompose $\boldsymbol{G}$ into $\boldsymbol{V}$ and a component orthogonal to $\boldsymbol{V}$ denoted $\boldsymbol{H}$,

$$\boldsymbol{G} = \boldsymbol{H} + \sigma_{\boldsymbol{G}}\boldsymbol{V}$$

where $\sigma_{\boldsymbol{G}} := \frac{\langle \boldsymbol{G}, \boldsymbol{V} \rangle}{\|\boldsymbol{V}\|_F^2}$, and under the assumption that $\boldsymbol{G}$ is centered, $\sigma_{\boldsymbol{G}} = \frac{\|\boldsymbol{G}\|_*}{n-1}$. Remarkably, the following lemma tells us that a *non-linear* function of $\boldsymbol{H}$ uniquely determines $\boldsymbol{G}$.

**Lemma 2.6.** *If $n > d + 1$, and $\boldsymbol{G}$ is rank $d$ and centered, then $-\sigma_{\boldsymbol{G}}$ is an eigenvalue of $\boldsymbol{H}$ with multiplicity $n - d - 1$. In addition, given another Gram matrix $\boldsymbol{G}'$ of rank $d'$, $\sigma_{\boldsymbol{G}'} - \sigma_{\boldsymbol{G}}$ is an eigenvalue of $\boldsymbol{H} - \boldsymbol{H}'$ with multiplicity at least $n - d - d' - 1$.*

*Proof.* Since $\boldsymbol{G}$ is centered, $\boldsymbol{1}_n \in \ker \boldsymbol{G}$, and in particular $\dim(\boldsymbol{1}_n^\perp \cap \ker \boldsymbol{G}) = n - d - 1$. If $x \in \boldsymbol{1}_n^\perp \cap \ker \boldsymbol{G}$, then

$$\boldsymbol{G}x = \boldsymbol{H}x + \sigma_{\boldsymbol{G}}\boldsymbol{V}x \Rightarrow \boldsymbol{H}x = -\sigma_{\boldsymbol{G}}x.$$

For the second statement, notice that $\dim(\boldsymbol{1}_n^\perp \cap \ker \boldsymbol{G} - \boldsymbol{G}') \ge n - d - d' - 1$. A similar argument then applies. $\qquad\square$

If $n > 2d$, then the multiplicity of the eigenvalue $-\sigma_{\boldsymbol{G}}$ is at least $n/2$. So we can trivially identify it from the spectrum of $\boldsymbol{H}$. This gives us a *non-linear* way to recover $\boldsymbol{G}$ from $\boldsymbol{H}$.

Now we can return to the task of recovering $\boldsymbol{K}^*$ from $\mathcal{M}(\widehat{\boldsymbol{K}})$. Indeed the above lemma implies that $\boldsymbol{G}^*$ (and hence $\boldsymbol{K}^*$ if $\boldsymbol{X}$ is full rank) can be recovered from $\boldsymbol{H}^*$ by computing an eigenvalue of $\boldsymbol{H}^*$. However $\boldsymbol{H}^*$ is recoverable from $\mathcal{L}(\boldsymbol{H}^*)$, which is itself well approximated by $\mathcal{L}(\widehat{\boldsymbol{H}}) = \mathcal{M}(\widehat{\boldsymbol{K}})$. The proof of the following theorem makes this argument precise.

**Theorem 2.7.** *Assume that $\boldsymbol{K}^*$ is rank $d$, $\widehat{\boldsymbol{K}}$ is rank $d'$, $n > d + d' + 1$, $\boldsymbol{X}$ is rank $p$ and $\boldsymbol{X}^T\boldsymbol{K}^*\boldsymbol{X}$ and $\boldsymbol{X}^T\widehat{\boldsymbol{K}}\boldsymbol{

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

# 4 Supplementary Materials

# 5 Proof of Results

## 5.1 Proof of Theorem 2.1

Our argument follows standard statistical learning theory techniques used in the classification literature. This framework is also similar to that used in the one bit matrix completion literature, see [12]. The main ingredient in the proof is the use of a Matrix Bernstein to bound the Rademacher complexity of our class.

By the Bounded Difference inequality,

$$
\begin{aligned}
R(\widehat{\boldsymbol{K}}) - R(\boldsymbol{K}^\star) &= \mathbb{R}(\widehat{\boldsymbol{K}}) - \widehat{R}(\widehat{\boldsymbol{K}}) + \widehat{R}(\widehat{\boldsymbol{K}}) - \widehat{R}(\boldsymbol{K}^\star) + \widehat{R}(\boldsymbol{K}^\star) - R(\boldsymbol{K}^\star) \\
&\leq 2 \sup_{\boldsymbol{K} \in \mathcal{K}_{\lambda,\gamma}} |\widehat{R}(\boldsymbol{K}) - R(\boldsymbol{K})| \\
&\leq 2\mathbb{E}[\sup_{\boldsymbol{K} \in \mathcal{K}_{\lambda,\gamma}} |\widehat{R}(\boldsymbol{K}) - R(\boldsymbol{K})|] + \sqrt{\frac{2\beta^2 \log 2/\delta}{|\mathcal{S}|}},
\end{aligned}
$$

where $\beta = \sup_{\boldsymbol{K} \in \mathcal{K}_{\lambda,\gamma}} |\ell((y_t \langle \boldsymbol{M}_t, \boldsymbol{K} \rangle) - \ell((y_{t'} \langle \boldsymbol{M}_{t'}, \boldsymbol{K} \rangle)| \leq L\gamma$ since $\langle \boldsymbol{M}_t, \boldsymbol{K} \rangle \leq \gamma$. Using standard symmetrization and contraction lemmas, we can introduce Rademacher random variables $\varepsilon_t \in \{-1, 1\}$ for all $t \in \mathcal{T}$ so that

$$
\begin{aligned}
\mathbb{E}\left[\sup_{\boldsymbol{K} \in \mathcal{K}_{\lambda,\gamma}} |\widehat{R}(\boldsymbol{K}) - R(\boldsymbol{K})|\right] &\leq \mathbb{E} \frac{2L}{|\mathcal{T}|} \sup_{\boldsymbol{K} \in \mathcal{K}_{\lambda,\gamma}} \left| \sum_{t \in \mathcal{S}} \varepsilon_t \langle \boldsymbol{M}_t, \boldsymbol{K} \rangle \right| \\
&\leq \mathbb{E} \frac{2L}{|\mathcal{S}|} \sup_{\boldsymbol{K} \in \mathcal{K}_{\lambda,\gamma}} \| \sum_{t \in \mathcal{S}} \varepsilon_t \boldsymbol{M}_t \| \|\boldsymbol{K}\|_* \\
&\leq \mathbb{E} \frac{2L\lambda}{|\mathcal{S}|} \sup_{\boldsymbol{K} \in \mathcal{K}_{\lambda,\gamma}} \| \sum_{t \in \mathcal{S}} \varepsilon_t \boldsymbol{M}_t \|
\end{aligned}
$$

We employ a matrix Bernstein bound, Theorem 6.6.1 in [13], to compute

$$
\mathbb{E}\| \sum_{t \in \mathcal{S}} \varepsilon_t \boldsymbol{M}_t \| \leq \sqrt{140 \frac{\|\boldsymbol{X}\boldsymbol{X}^T\|}{n} |\mathcal{S}| \log p} + 2 \log p.
$$

To see this, it suffices to bound $\left\| \sum_{t \in \mathcal{T}} \boldsymbol{M}_t^2 \right\|$ which is done in Lemma 5.1. Plugging this in above gives

$$
\mathbb{E} \frac{2L\lambda}{|\mathcal{S}|} \left\| \sum_{t \in \mathcal{S}} \varepsilon_t \boldsymbol{M}_t \right\| \leq 2L \left( \sqrt{\frac{140\lambda^2 \frac{\|\boldsymbol{X}\boldsymbol{X}^T\|}{n} \log p}{|\mathcal{S}|}} + \frac{2 \log p}{|\mathcal{S}|} \right)
$$

**Lemma 5.1.**

$$
\frac{1}{n\binom{n-1}{2}} \left\| \sum_{t \in \mathcal{T}} \boldsymbol{M}_t^2 \right\| \leq 70 \frac{\|\boldsymbol{X}\boldsymbol{X}^T\|}{n}
$$

*Proof.* Let $\boldsymbol{e}_i$ be the $i^{th}$ standard basis vector. For a triplet $t = (i, j, k)$, define

$$
\boldsymbol{L}_t = \boldsymbol{e}_i \boldsymbol{e}_k^T + \boldsymbol{e}_k \boldsymbol{e}_i^T - \boldsymbol{e}_i \boldsymbol{e}_j^T - \boldsymbol{e}_j \boldsymbol{e}_i^T + \boldsymbol{e}_j \boldsymbol{e}_j^T - \boldsymbol{e}_k \boldsymbol{e}_k^T
$$

(in particular $\boldsymbol{L}_t$ is the matrix corresponding to the operator $\boldsymbol{L}_t$ given in Section 2.3). A computation shows that $\langle \boldsymbol{L}_t, \boldsymbol{X}^T \boldsymbol{K} \boldsymbol{X} \rangle = \langle \boldsymbol{M}_t, \boldsymbol{K} \rangle$ and moreover $\boldsymbol{M}_t = \boldsymbol{X} \boldsymbol{L}_t \boldsymbol{X}^T$. By definition,

$$
\begin{aligned}
\sum_{t \in \mathcal{T}} \boldsymbol{M}_t^2 &= \sum_{t \in \mathcal{T}} \boldsymbol{X} \boldsymbol{L}_t \boldsymbol{X}^T \boldsymbol{X} \boldsymbol{L}_t \boldsymbol{X} \\
&= \boldsymbol{X} \left( \sum_{t \in \mathcal{T}} \boldsymbol{L}_t \boldsymbol{X}^T \boldsymbol{X} \boldsymbol{L}_t \right) \boldsymbol{X}^T
\end{aligned}
$$

We now focus our attention on simplifying the middle term. Firstly, note that we can assume that the $X$'s are centered, i.e. $X\mathbf{1}_n = 0$. To see this, note that the $L_t$'s are centered so in particular, $L_t V = L_t$. Then

$$L_t X^T X L_t = L_t V^T X^T X V L_t = L_t (XV)^T (XV) L_T$$

so we can replace $X$ with $XV$, i.e. we can center $X$. Also note that note that centering $X$ only diminishes the operator norm $XX^T$, so centering does not affect the statement of the bound, and furthermore a tighter statement is certainly possible by assuming that $X$ is centered.

Using the reduction to a centered $X$, a computation (omitted due to length) shows that

$$\left(\sum_{t\in T} L_t X^T X L_t\right)_{i,j} = \begin{cases} (2n-3)\|X^T X\|_* + (n^2 - 3n)\|x_i\|^2 & i = j \\ (n-4)\langle x_i, x_j\rangle - (n-2)\|x_j\|^2 - (n-2)\|x_i\|^2 - \|X^T X\|_* & i \neq j \end{cases}$$

To bound $\|\sum_{t\in\mathcal{T}} L_t X^T X L_t\| \leq 7n^2$, by Gershgorin's Circle Theorem we just have to bound the sums of the absolute values of the entries in each row. This ends up being,

$$(2n-3+n-1)\|X^T X\|_* + (n^2 - 3n + (n-1)(n-2))\|x_i\|^2 + (n-2)\sum_{i\neq j}\|x_j\|^2$$
$$+ (n-4)\sum_{i\neq j}|\langle x_i, x_j\rangle|$$
$$\leq (2n-3+n-1+n-2)\|X^T X\|_* + (n^2 - 3n + (n-1)(n-2) - 2)\|x_i\|^2$$
$$+ (n-4)\sum_j \|x_i\|\|x_j\|$$
$$\leq (4n-6)\|X^T X\|_* + (2n^2 - 6n)\|x_i\|^2 + n(n-4)\max_j \|x_j\|^2$$
$$\leq (4n-6)n\max_j \|x_j\|^2 + (2n^2 - 6n)\max_j \|x_j\|^2 + (n^2 - 4n)\max_j \|x_j\|^2$$
$$\leq 7n^2 \max_j \|x_j\|^2$$

So $\|\sum_{t\in\mathcal{T}} L_t X^T X L_t\| \leq 7n^2$ and

$$\frac{1}{n\binom{n-1}{2}}\left\|\sum_{t\in\mathcal{T}} X L_t X^T X L_t X\right\| \leq 70\frac{\|XX^T\|}{n}$$

using the fact that $\frac{2n^2}{(n-1)(n-2)} \leq 10$ for positive $n \geq 3$. $\qquad\square$

## 5.2 Proof of Theorem 2.3

We will need the following lemma relating the KL-divergence to squared distance in this section and in the proof of Theorem 2.7.

**Lemma 5.2.** *Let $y, z \in (0,1)$, then*

$$2(z-y)^2 \leq KL(z||y) \leq \frac{(z-y)^2/2}{\inf_{x\in(0,1)} x(1-x)}$$

*Proof.* For $y, z \in (0,1)$ let $g(z) = z\log\frac{z}{y} + (1-z)\log\frac{1-z}{1-y}$. Then $g'(z) = \log\frac{z}{1-z} - \log\frac{y}{1-y}$ and $g''(z) = \frac{1}{z(1-z)}$. By Taylor's theorem, for some $\eta$ in the interval between $y$ and $z$, $g(z) = \frac{g''(\eta)}{2}(z-y)^2$. So for a lower bound,

$$g(z) \geq \frac{(z-y)^2/2}{\sup_{x\in(0,1)} x(1-x)} \geq 2(z-y)^2.$$

Similarly an upper bound is given by,

$$g(z) \leq \frac{(z-y)^2/2}{\inf_{x \in (0,1)} x(1-x)}$$

$\square$

Now we resume the proof of Theorem 2.3. Fix $\boldsymbol{X} = \boldsymbol{I}$. Given triplet comparisons generated according to $\boldsymbol{K}$, we are interested in finding the minimax lower bound,

$$\inf_{\widehat{\boldsymbol{K}}} \sup_{\boldsymbol{K} \in \mathcal{K}'_{\lambda,\gamma}} \mathbb{E}[R(\widehat{\boldsymbol{K}})] - R(\boldsymbol{K})$$

Where as previously computed in Section 2.3

$$R(\widehat{\boldsymbol{K}}) - R(\boldsymbol{K}) = \frac{1}{|\mathcal{T}|} \sum_{t \in \mathcal{T}} f(\langle \boldsymbol{M}_t, \boldsymbol{K} \rangle) \log \frac{f(\langle \boldsymbol{M}_t, \boldsymbol{K} \rangle)}{f(\langle \boldsymbol{M}_t, \widehat{\boldsymbol{K}} \rangle)} + (1 - f(\langle \boldsymbol{M}_t, \boldsymbol{K} \rangle)) \log \frac{1 - f(\langle \boldsymbol{M}_t, \boldsymbol{K} \rangle)}{1 - f(\langle \boldsymbol{M}_t, \widehat{\boldsymbol{K}} \rangle)}$$

Lemma 2.4 implies,

$$R(\widehat{\boldsymbol{K}}) - R(\boldsymbol{K}) \geq \frac{2C_f^2}{|\mathcal{T}|} \|\mathcal{M}(\boldsymbol{K}) - \mathcal{M}(\widehat{\boldsymbol{K}})\|_2^2.$$

where $C_f = \inf_{|x| \leq \gamma} f'(x)$. We will construct a set $\kappa \subset \mathcal{K}'_{\lambda,\gamma}$ so that for any two $\boldsymbol{K}^1, \boldsymbol{K}^2 \in \kappa$, with $K_1 \neq K_2$,

- $\frac{2C_f^2}{|\mathcal{T}|} \|\mathcal{M}(\boldsymbol{K}^1) - \mathcal{M}(\boldsymbol{K}^2)\|_F^2 \geq 4s_n^2$, for $\boldsymbol{K}^1 \neq \boldsymbol{K}^2$

- Let $P_{\boldsymbol{K}}^{\mathcal{S}}$ denote the sample distribution of a set of $|\mathcal{S}|$ samples conditioned on it being drawn from $\boldsymbol{K} \in \kappa$. Then we also require $KL(P_{\boldsymbol{K}^1}^{\mathcal{S}} || P_{\boldsymbol{K}^2}^{\mathcal{S}}) \leq \frac{1}{16} \ln |\kappa|$

Following an argument similar to the proof of Theorem 2 in [14], it will then follow from a variant of Fano's inequality, namely Lemma A.1 from [15], that

$$\inf_{\widehat{\boldsymbol{K}}} \sup_{\boldsymbol{K} \in \mathcal{K}'_{\lambda,\gamma}} \mathbb{E}[R(\widehat{\boldsymbol{K}})] - R(\boldsymbol{K}) \geq s_n^2.$$

By Lemma 8.3 of [16], there exists a subset $\kappa \subset \mathcal{K}'_{\lambda,\gamma}$, and an absolute constant $0 < C_1 < 1$ such that

- $\ln |\kappa| \geq C_1 d \ln \frac{p}{d}$
- Each element of $\kappa$ has sparsity $d$, is 0 away from the diagonal, and on the diagonal the elements are either 0 or $\gamma$, for a value of $\gamma \geq 0$ we will choose later.
- For all $\boldsymbol{K}^i, \boldsymbol{K}^j \in \kappa$, $\|\boldsymbol{K}^i - \boldsymbol{K}^j\|_0 \geq C_1 d$.

Therefore, for $\boldsymbol{K}^1, \boldsymbol{K}^2 \in \kappa$, we need only to show $KL(\boldsymbol{K}^1 || \boldsymbol{K}^2) \leq \frac{1}{16} \ln |\kappa|$. Using the fact that $\boldsymbol{X} = \boldsymbol{I}$,

$$\frac{2C_f^2}{|\mathcal{T}|} \|\mathcal{M}(\boldsymbol{K}^1) - \mathcal{M}(\boldsymbol{K}^2)\|_2^2 \geq \frac{2C_f^2}{|\mathcal{T}|} p \sum_{j<k} ((\boldsymbol{K}_{kk}^1 - \boldsymbol{K}_{kk}^2) - (\boldsymbol{K}_{jj}^1 - \boldsymbol{K}_{jj}^2))^2$$

$$\geq \frac{2C_f^2 C_1 p d (p - 2d) \gamma^2}{|\mathcal{T}|}$$

To see the second to last inequality, note that there are at least $C_1 d(p - 2d)$ pairs of indices $j, k$ where $\boldsymbol{K}_{kk}^1 \neq \boldsymbol{K}_{kk}^2$ but $\boldsymbol{K}_{jj}^1 = \boldsymbol{K}_{jj}^2$, because $\boldsymbol{K}^1$ and $\boldsymbol{K}^2$ share at least $p - 2d$ entries on their diagonal that are both 0. Each such entry contributes a $\gamma^2$ to the sum.

In particular choose,

$$s_n^2 = \frac{C_f^2 C_1 p d (p - 2d) \gamma^2}{2|\mathcal{T}|}.$$

We proceed by selecting $\gamma$ such that $KL(P_{\boldsymbol{K}^1}^{\mathcal{S}}||P_{\boldsymbol{K}^2}^{\mathcal{S}}) \leq \frac{1}{16} \ln |\kappa|$. Assume our samples are $\mathcal{S} = \{(t, y_t)\}$. Then since the samples are i.i.d.

$$KL(P_{\boldsymbol{K}^1}^{\mathcal{S}}||P_{\boldsymbol{K}^2}^{\mathcal{S}}) = \sum_{t \in \mathcal{S}} KL(P_{\boldsymbol{K}^1}(t)||P_{\boldsymbol{K}^2}(t))$$

where $P_{\boldsymbol{K}^i}(t)$ is the distribution of $y_t$ conditioned on $\boldsymbol{K}^i$, in particular the probability of $y_t = -1$ is $f(\langle \boldsymbol{M}_t, \boldsymbol{K}^i \rangle)$.

We can bound each term of the sum above using the upper bound from Lemma 5.2.

$$
\begin{aligned}
KL(P_{\boldsymbol{K}^1}(t)||P_{\boldsymbol{K}^2}(t)) &\leq \frac{(f(\langle M_t, \boldsymbol{K}^1 \rangle) - f(\langle M_t, \boldsymbol{K}^2 \rangle))^2}{2 \inf_t f(\langle M_t, \boldsymbol{K}^2 \rangle)(1 - f(\langle M_t, \boldsymbol{K}^2 \rangle))} \\
&\leq \frac{(\langle M_t, \boldsymbol{K}^1 - \boldsymbol{K}^2 \rangle)^2 \sup_{|\nu| \leq \gamma} f'(\nu)^2}{2 \inf_{|x| \leq \gamma} f(x)(1 - f(x))} \\
&\leq \frac{\gamma^2 \sup_{|\nu| \leq \gamma} f'(\nu)^2}{2 \inf_{|x| \leq \gamma} f(x)(1 - f(x))}
\end{aligned}
$$

Summing over $t \in \mathcal{S}$, we require that

$$KL(P_{\boldsymbol{K}^1}^{\mathcal{S}}||P_{\boldsymbol{K}^2}^{\mathcal{S}}) \leq \frac{\gamma^2 |\mathcal{S}| \sup_{|\nu| \leq \gamma} f'(\nu)^2}{2 \inf_{|x| \leq \gamma} f(x)(1 - f(x))} \leq \frac{C_1}{16} d \ln \frac{p}{d} \leq \frac{1}{16} \ln |\kappa|,$$

so in particular, we will take

$$\frac{\gamma^2 \sup_{|\nu| \leq \gamma} f'(\nu)^2}{2 \inf_{|x| \leq \gamma} f(x)(1 - f(x))} = \frac{C_1}{16|\mathcal{S}|} d \ln \frac{p}{d}$$

From this point on, let's take $\lambda = p$, and $d = \frac{p}{4}$. Now we have a few additional constraints on $\gamma$,

- Since $\|\boldsymbol{K}^i\|_{1,2} \leq \lambda$ for each $\boldsymbol{K}^i \in \kappa$, we require $\gamma d \leq \lambda$, so in particular $\gamma \leq 4$.
- In addition, we are going to require $\gamma \geq 1$ since we will need $p\gamma \geq \lambda$ (used below).

Based on these conditions, we just take $\gamma = 2$ and after simplification choose,

$$|S| := \frac{C_1 p \ln 4 \inf_{|x| \leq \gamma} f(x)(1 - f(x))}{32 \gamma^2 \sup_{|\nu| \leq \gamma} f'(\nu)^2}$$

Now we are finally in a position to use our choice of $\gamma, d, \lambda$ and $|\mathcal{S}|$. We see that

$$
\begin{aligned}
s_n^2 = \frac{C_f^2 C_1 p d(p - 2d)\gamma^2}{2|\mathcal{T}|} &= \frac{C_f^2 C_1 p^2 \gamma \lambda}{16|\mathcal{T}|} \qquad \text{(since } p\gamma \geq \lambda) \\
&\geq \frac{C_f^2 C_1 \gamma \lambda}{8p} \\
&\geq \frac{C_f^2 \sqrt{\inf_{|x| \leq \gamma} f(x)(1 - f(x))}}{8p \sqrt{\sup_{|\nu| \leq 2} f'(\nu)^2}} \sqrt{\frac{C_1^3 \ln 4}{32} \frac{p}{|\mathcal{S}|}} \\
&= \frac{C_f^2}{32} \sqrt{\frac{\inf_{|x| \leq \gamma} f(x)(1 - f(x))}{\sup_{|\nu| \leq \gamma} f'(\nu)^2}} \sqrt{\frac{C_1^3 \ln 4}{2} \lambda^2 \frac{\|\boldsymbol{X}\boldsymbol{X}^T\|}{n}}{|\mathcal{S}|}
\end{aligned}
$$

where the final equality follows from the fact that we have chosen $\boldsymbol{X} = \boldsymbol{I}_n$ so $n = p$. $\qquad \square$

## 5.3 Proof of Lemma 2.4

*Proof of Lemma 2.4.* As computed prior to the statement of Theorem 2.7.

$$R(\widehat{\boldsymbol{K}}) - R(\boldsymbol{K}^*) = \frac{1}{|\mathcal{T}|} \sum_{t \in \mathcal{T}} KL(f(\langle \boldsymbol{M}_t, \boldsymbol{K}^* \rangle)||f(\langle \boldsymbol{M}_t, \widehat{\boldsymbol{K}} \rangle))$$

Now using Lemma 5.2 with $z = f(\langle \boldsymbol{M}_t, \boldsymbol{K}^* \rangle)$ and $y = f(\langle \boldsymbol{M}_t, \widehat{\boldsymbol{K}} \rangle)$ we see

$$KL(f(\langle \boldsymbol{M}_t, \boldsymbol{K}^* \rangle) \| f(\langle \boldsymbol{M}_t, \widehat{\boldsymbol{K}} \rangle)) \geq 2C_f^2 (\langle \boldsymbol{M}_t, \boldsymbol{K}^* \rangle - \langle \boldsymbol{M}_t, \widehat{\boldsymbol{K}} \rangle)^2$$

Summing over all $t \in \mathcal{T}$

$$
\begin{aligned}
R(\widehat{\boldsymbol{K}}) - R(\boldsymbol{K}^\star) &\geq \frac{2C_f^2}{|\mathcal{T}|} \sum_{t \in T} (\langle \boldsymbol{M}_t, \boldsymbol{K}^* \rangle - \langle \boldsymbol{M}_t, \widehat{\boldsymbol{K}} \rangle)^2 \\
&= \frac{2C_f^2}{|\mathcal{T}|} \sum_{t \in T} (\langle \boldsymbol{M}_t, \widehat{\boldsymbol{K}} - \boldsymbol{K}^\star \rangle)^2 = \frac{2C_f^2}{|\mathcal{T}|} \| \mathcal{M}(\widehat{\boldsymbol{K}}) - \mathcal{M}(\boldsymbol{K}^*) \|_2^2.
\end{aligned}
$$

$\square$

## 5.4    Proof of Theorem 2.7

Before launching into the proof of Theorem 2.7, we first prove an auxiliary set of results that depend on the classical correspondence between centered Gram matrices and Euclidean distance matrices. For a more in depth discussion of this correspondence, we refer interested readers to [17]. Let $\mathbb{S}_h^n$ be the subspace of symmetric hollow matrices, i.e. symmetric matrices with zero diagonal, and let $\mathbb{S}_c^n$ be the subspace of centered Gram matrices, i.e. positive semi-definite matrices with $\boldsymbol{1}_n$ in their kernel.

Note that $\dim \mathbb{S}_h^n = \dim \mathbb{S}_c^n = \binom{n}{2}$. In fact these spaces are isomorphic with an explicit linear isomorphism given by the maps

$$\mathbb{S}_h^n \to \mathbb{S}_c^n : \boldsymbol{D} \to -\frac{1}{2} \boldsymbol{V} \boldsymbol{D} \boldsymbol{V}$$

with inverse

$$\mathbb{S}_c^n \to \mathbb{S}_h^n : \boldsymbol{G} \to \text{diag}(\boldsymbol{G}) \boldsymbol{1}_n^T - 2\boldsymbol{G} + \boldsymbol{1}_n \text{diag}(\boldsymbol{G})^T$$

where again, $\boldsymbol{V} = \boldsymbol{I}_n - \frac{1}{n} \boldsymbol{1}_n \boldsymbol{1}_n^T$.

Given a set of centered points $\boldsymbol{X} \in \mathbb{R}^p$, then under the isomorphism above, the associated Gram matrix $\boldsymbol{G} \in \mathbb{S}_c^n$ maps to the squared distance matrix $\boldsymbol{D} \in \mathbb{S}_h^n$. In particular, a matrix in $\mathbb{S}_h^n$ is a valid Euclidean distance matrix if and only if $-\frac{1}{2} \boldsymbol{V} \boldsymbol{D} \boldsymbol{V}$ is a centered Gram matrix.

Given a triplet $t = (i, j, k) \in \mathcal{T}$, we can define an operator $\Delta_t(\boldsymbol{D}) := \boldsymbol{D}_{ij} - \boldsymbol{D}_{ik}$ and

$$\Delta(\boldsymbol{D}) := (\Delta_t(\boldsymbol{D}) | \text{ for } t \in \mathcal{T})$$

analogous to $\mathcal{L}$ and $\mathcal{M}$. In particular, for associated $\boldsymbol{D}$ and $\boldsymbol{G}$, $\Delta_t(\boldsymbol{D}) = \mathcal{L}_t(\boldsymbol{G})$ for all $t$ so $\Delta(\boldsymbol{D}) = \mathcal{L}(\boldsymbol{G})$. We can now prove the key lemmas used in the proof of 2.7.

**Lemma 5.3.** *The null space of $\mathcal{L}$ is one dimensional, spanned by $\boldsymbol{V}$.*

*Proof.* Lemma 2 in [11] shows $\ker \Delta$ is one dimensional and is spanned by $\boldsymbol{J} = \boldsymbol{1}_n \boldsymbol{1}_n^T - \boldsymbol{I}_n$. A computation shows that $-\frac{1}{2} \boldsymbol{V} \boldsymbol{J} \boldsymbol{V} = \frac{1}{2} \boldsymbol{V}$. Since $\mathcal{L}(\boldsymbol{V}) = \Delta(\boldsymbol{J}) = \boldsymbol{0}$, $\boldsymbol{V}$ spans $\ker \mathcal{L}$. $\square$

We rely on an analogous statement for distance matrices given in Lemma 3 in [11].

**Lemma 5.4.** *Let $\boldsymbol{G} \in \mathbb{S}_c^n$ and $\boldsymbol{H}$ the component of $\boldsymbol{G}$ orthogonal $\boldsymbol{V}$ then $\| \mathcal{L}(\boldsymbol{H}) \|^2 \geq n \| \boldsymbol{H} \|_F^2$.*

*Proof.* Again, let $\boldsymbol{D}$ be the symmetric hollow matrix corresponding to $\boldsymbol{G}$. We can take a decomposition of $\boldsymbol{D}$ into a component perpendicular to $\ker \Delta$

$$\boldsymbol{D} = \boldsymbol{C} + \sigma_D \boldsymbol{J}.$$

Applying $-\frac{1}{2} \boldsymbol{V} \cdot \boldsymbol{V}$ to both sides we get,

$$\boldsymbol{G} = -\frac{1}{2} \boldsymbol{V} \boldsymbol{C} \boldsymbol{V} + \frac{\sigma_D}{2} \boldsymbol{V}.$$

We claim that $\boldsymbol{H} = -\frac{1}{2} \boldsymbol{V} \boldsymbol{C} \boldsymbol{V}$ and $\sigma_{\boldsymbol{G}} = \sigma_D / 2$. It suffices to prove that $\boldsymbol{V} \boldsymbol{C} \boldsymbol{V}$ is perpendicular to $\boldsymbol{V}$. To see this note that $\langle \boldsymbol{V} \boldsymbol{C} \boldsymbol{V}, \boldsymbol{V} \rangle = \langle \boldsymbol{C}, \boldsymbol{V} \rangle = 0$, since $\boldsymbol{C}$ is hollow and perpendicular to $\boldsymbol{J}$.

We now apply Lemma 3 in [11] which shows that the minimal eigenvalue of $\Delta$ is $n$.

$$
\begin{aligned}
\|\mathcal{L}(\boldsymbol{H})\|^2 &= \|\Delta(\boldsymbol{C})\|^2 \\
&\geq n\|\boldsymbol{C}\|_F^2 && \text{(since } C \text{ is perpendicular to the kernel of } \Delta) \\
&\geq n\left\|-\frac{1}{2}\boldsymbol{V}\boldsymbol{C}\boldsymbol{V}\right\|_F^2 && \text{(Since } \boldsymbol{V} \text{ is a projection.)} \\
&\geq n\|\boldsymbol{H}\|_F^2
\end{aligned}
$$

$\square$

*Proof of Theorem 2.7.* We begin by applying Lemma 2.6 in the specific case where $\boldsymbol{G}^* = \boldsymbol{X}^T \boldsymbol{K}^* \boldsymbol{X}$ and $\widehat{\boldsymbol{G}} = \boldsymbol{X}^T \widehat{\boldsymbol{K}} \boldsymbol{X}$ with $\boldsymbol{H}^*$ and $\widehat{\boldsymbol{H}}$ defined analogously to above. Firstly, by definition

$$
\widehat{\boldsymbol{G}} - \boldsymbol{G}^* = \widehat{\boldsymbol{H}} - \boldsymbol{H}^* + (\sigma_{\widehat{\boldsymbol{G}}} - \sigma_{\boldsymbol{G}^*})\boldsymbol{V}
$$

By orthogonality

$$
\begin{aligned}
\|\widehat{\boldsymbol{G}} - \boldsymbol{G}^*\|_F^2 &= \|\widehat{\boldsymbol{H}} - \boldsymbol{H}^*\|_F^2 + (\sigma_{\widehat{\boldsymbol{G}}} - \sigma_{\boldsymbol{G}^*})^2\|\boldsymbol{V}\|_F^2 \\
&= \|\widehat{\boldsymbol{H}} - \boldsymbol{H}^*\|_F^2 + (n-1)(\sigma_{\widehat{\boldsymbol{G}}} - \sigma_{\boldsymbol{G}^*})^2 && (\text{Since } \|\boldsymbol{V}\|_F^2 = n-1) \\
&\leq \|\widehat{\boldsymbol{H}} - \boldsymbol{H}^*\|_F^2 + \frac{n-1}{(n-d-d'-1)}\|\widehat{\boldsymbol{H}} - \boldsymbol{H}^*\|_F^2
\end{aligned}
$$

(By Lemma 2.6 $\sigma_{\widehat{\boldsymbol{G}}} - \sigma_{\boldsymbol{G}^*}$ is a repeated eigenvalue with multiplicity $n - d - d' - 1$)

$$
= C_{d,d'}\|\widehat{\boldsymbol{H}} - \boldsymbol{H}^*\|_F^2.
$$

Now,

$$
\begin{aligned}
\|\mathcal{M}(\widehat{\boldsymbol{K}}) - \mathcal{M}(\boldsymbol{K}^*)\|_2^2 &= \|\mathcal{L}(\boldsymbol{X}^T \boldsymbol{K} \boldsymbol{X}) - \mathcal{L}(\boldsymbol{X}^T \boldsymbol{K}^* \boldsymbol{X})\|^2 \\
&\geq n\|\widehat{\boldsymbol{H}} - \boldsymbol{H}^*\|_F^2 && \text{(Using Lemma 5.4)} \\
&\geq \|\widehat{\boldsymbol{G}} - \boldsymbol{G}^*\|_F^2 && \text{(From the above.)} \\
&= \frac{n}{C_{d,d'}}\|\boldsymbol{X}^T \widehat{\boldsymbol{K}} \boldsymbol{X} - \boldsymbol{X}^T \boldsymbol{K}^* \boldsymbol{X}\|_F^2 \\
&\geq \frac{n\sigma_{\min}(\boldsymbol{X}\boldsymbol{X}^T)^2}{C_{d,d'}}\|\widehat{\boldsymbol{K}} - \boldsymbol{K}^*\|_F^2
\end{aligned}
$$

To see the last line, recall $\text{vec}(\boldsymbol{X}^T \boldsymbol{K} \boldsymbol{X}) = (\boldsymbol{X}^T \otimes \boldsymbol{X}^T)\text{vec}(\boldsymbol{K})$. Now, the minimal eigenvalue of $\boldsymbol{X}^T \otimes \boldsymbol{X}^T$ is $\sigma_{\min}(\boldsymbol{X}\boldsymbol{X}^T)$ which is nonzero since $\boldsymbol{X}$ is rank $p$.

So we see from Lemma 2.4, that

$$
\frac{n\sigma_{\min}(\boldsymbol{X}\boldsymbol{X}^T)^2}{|\mathcal{T}|}\|\boldsymbol{K} - \widehat{\boldsymbol{K}}\|_F^2 \leq \frac{C_{d,d'}}{C_f^2}(R(\widehat{\boldsymbol{K}}) - R(\boldsymbol{K}^*))
$$

The result now follows from Theorem 2.1.

$\square$