[Reviews · NeurIPS 2017]

Reviewer 1



The paper presents theoretical work towards learning low-dim/low rank and sparse metrics, deriving the minimax bounds on the generalization error and shows sample complexity based on ‘d' and ‘p'. The paper is well written and revolves around/builds upon \ref 2 and \ref 3 whilst filling the gaps and providing more in-depth theoretical justifications specific to low-rank and sparse metrics. Section 2.3 provides identification bounds but what are the authors thoughts on trying out, say for example, the Bregman divergence for comparing K and K^*?

Reviewer 2



Summary of the paper: The paper considers the problem of learning a low-rank / sparse and low-rank Mahalanobis distance under relative constraints dist(x_i,x_j) < dist(x_i,x_k) in the framework of regularized empirical risk minimization using trace norm / l_{1,2} norm regularization. The contributions are three theorems that provide 1) an upper bound on the estimation error of the empirical risk minimizer 2) a minimax lower bound on the error under the log loss function associated to the model, showing near-optimality of empirical risk minimization in this case 3) an upper bound on the deviation of the learned Mahalanobis distance from the true one (in terms of Frobenius norm) under the log loss function associated to the model. Quality: My main concern here is the close resemblance to Jain et al. (2016). Big parts of the present paper have a one-to-one correspondence to parts in that paper. Jain et al. study the problem of learning a Gram matrix G given relative constraints dist(x_i,x_j) < dist(x_i,x_k), but without being given the coordinates of the points x_i (ordinal embedding problem). The authors do a good job in explaining that the ordinal embedding problem is a special case of the considered metric learning problem (Section 2.4). However, I am wondering whether such a relationship in some sense holds the other way round too and how it could be exploited to derive the results in the present paper from the results in Jain et al. (2016) without repeating so many things: can’t we exploit the relationship G=X^T K X to learn K from G? At least in the noise-free case, if G=Y^T Y we can simply set K=L^T L, where L solves Y^T = X^T L^T? I did not check the proofs in detail, but what I checked is correct. Also, many things can be found in Jain et al. (2016) already. In Line 16, d_K(x_i,x_j) is defined as (x_i-x_j)^T K (x_i-x_j), but this is not a metric. To be rigorous, one has to take the square root in this definition and replace d_K(x_i,x_j) by d_K(x_i,x_j)^2 in the following. In Corollary 2.1.1, the assumption is max_i \|x_i\|^2 = O(\sqrt{d}\log n), but in Theorem 2.1 it is max_i \|x_i\|^2 = 1. How can we have a weaker assumption in the corollary than in the theorem? There is a number of minor things that should be corrected, see below. Clarity: At some points, the presentation of the paper could be improved: Apparently, in Theorem 2.3 the set K_{\lambda,\gamma} corresponds to l_{1,2} norm regularization, but the definition from Line 106 has never been changed. That’s quite confusing. Line 76: “a noisy indication of the triplet constraint d_{\hat{K}}(x_i,x_j) < d_{\hat{K}}(x_i,x_k)” --- in the rest of the paper, hat{K} denotes the estimate? And, if I understand correctly, we have y_t=-1 if the constraint is true. From the current formulation one would expect y_t=+1 in this case. Originality: I am missing a comparison to closely related results in the literature. What is the difference between Theorem 2.1 and Inequality (5) in combination with Example 6 in [3] (Bellet et al., 2015---I refer to the latest arxiv version)? The authors claim that “past generalization error bounds ... failed to quantify the precise dependence on ... dimension”. However, the dimension p only appears via log p and is not present in the lower bound anymore, so it’s unclear whether the provided upper bound really quantifies the precise dependence. Huang et al. (2011) is cited incorrectly: it DOES consider relative constraints and does NOT use Rademacher complexities. Significance: In the context of metric learning, the results might be interesting, but a big part of the results (formulated in the setting of ordinal embedding --- see above) and techniques can be found already in Jain et al. (2016). Minor points: - Line 75, definition of T: in a triple (i,j,k), we assume j < k - Line 88: that’s an equation and not a definition - Line 98: if I_n is used to denote the identity, why not use 1_n? - Line 99: XV instead of VX - Line 100/101: also define the K-norm used in Line 126 and the ordinary norm as Euclidean norm or operator norm - Be consistent with the names of the norm: in Line 101 you say “the mixed 1,2 norm”, in Line 144 “a mixed l_{1,2} norm”. - In Corollary 2.1.1: shouldn’t the log n-term be squared? - Line 190: remove the dot - Line 212: I think “rank of the” is missing - Typos in Line 288, Line 372, Line 306, Line 229, Line 237, Line 240, Line 259 - Line 394: “We” instead of “I” - Line 335: for K^1 \noteq K^2

Reviewer 3



In this paper, the author has investigated the learning of low-dimensional metrics. In general, the technique of this paper is sound and the paper is well written. My concern is mainly focused on the experimental part. (1) The author adopt the simulation dataset for performance comparison. What about the performance on large-scale real datasets, e.g., image, text. (2) Nuclear norm and L_{1,2} norm are adopted for comparison. To validate the effectiveness, the author should list more state-of-the-art methods here and compare their performance from a different perspective. (3) The conclusion part is missing. The abstract part as well as the contributions claimed in part 1 need improvement. The author should clear state the contributions of paper and convincingly demonstrate how each of these contributions advance the state-of-the-art.